# Evaluation of the measurement properties of intimate partner violence screening instruments for the general population: A COSMIN-based international systematic review

Yanjia Li[1☉], Guiyun Wang[2☉], Jiarui Chen[1,3,4]*, Qing Xia[3], Keyi Chen[3], Suqi Ou[1], Siyuan Tang[1,4]

1 Xiangya School of Nursing, Central South University, Changsha, China, 2 Shandong Xiehe University, Jinan, China, 3 Changsha Medical College, Changsha, China, 4 Xiangya Center for Evidence-Based Nursing Practice & Healthcare Innovation: A JBI Centre of Excellence, Changsha, China

☉ These authors contributed equally to this work.
* chenjr@csu.edu.cn

**Data Availability Statement:** All relevant data are within the paper and its Supporting information files.

## Abstract

### Aim

To systematically appraise, compare, and summarize the measurement properties of intimate partner violence screening instruments for the general population and provide recommendations.

### Methods

We searched PubMed, Embase, Web of Science, ProQuest Dissertations & Theses Global and EBSCO Psychology Behavioral Sciences Collection from their establishment to March 2024 using systematic search strategies. The methodological quality of the instruments that met the inclusion criteria and their measurement properties were assessed using the COSMIN methodology, and the COSMIN recommendations were followed. We reported this study using PRISMA 2020 checklist.

### Results

A total of 23 studies were eventually included, and 18 instruments were identified. The evaluation of the methodological quality indicates poor content validity but good structural validity; however, that of the criterion validity in most of the studies was inadequate. Measurement error and responsiveness were not assessed. Four screening instruments could be used in the interim. The remaining 14 instruments were not recommended for use.

### Conclusions

The overall methodological quality of most of the assessed instruments was insufficient. A rigorous intimate partner violence screening instrument with good measurement properties

**Funding:** This research was funded by the Natural Science Foundation of Hunan Province (grant number 2022JJ40641) and the Changsha Natural Science Foundation (grant number kq2202114). The funders had no role in the methods, data extraction, data synthesis, or preparation of the manuscript.

**Competing interests:** No conflict of interest has been declared by the authors.

is urgently required to identify and screen for intimate partner violence in the general population.

## PROSPERO number

CRD42022365247.

## Introduction

Intimate partner violence (IPV) is a major public health issue that affects the human rights of people in various cultures and societies [1–3]. IPV refers to the infliction of physical, psychological, or sexual harm by a partner or ex-partner [4]. It encompasses physical violence, sexual violence, emotional-psychological abuse, and controlling behaviors according to the World Health Organization [5]. Intimate partners may include couples, ex-couples, boyfriends/girlfriends, dating partners, sexual partners, or other romantic type of relationship. They can also be current or former partners, of the same or opposite sex, and may or may not be cohabiting [6]. The term "domestic violence" is a broader term that contains the abuse of children or the elderly or the abuse by any member from the family, while it was used in many countries to refer to intimate partner violence [5]. In this study, we therefore conducted a database search using the term "Intimate Partner Violence."

In a study by the World Health Organization, 27% of women aged 15–49 years had been subjected to physical and/or sexual violence by their partners, and 13% of them had experienced such events in the previous 12-month period [7]. IPV can result in negative physical and emotional outcomes for the people who experience violence and their families [2]. The physical health consequences include acute illnesses and chronic and pain-based disorders, such as respiratory, urinary tract, and sexually transmitted illnesses, insomnia, headache, menstrual-related disorders, pelvic pain, and functional gastrointestinal and reproductive diseases [8–10]. Mental health issues are also reported among people who are experiencing or have experienced IPV, and these can have potentially serious consequences. Anxiety, depression, posttraumatic stress disorder, and suicide are all common among this population [11, 12]. Importantly, the severity of these mental health issues tends to escalate with the severity of the IPV experienced.

In 2009, a systematic review [13] summarized existing psychometric data of four IPV screening instruments in healthcare settings based on the U.S. Preventive Services Task Force criteria. Whereas results showed the psychometric properties of all screening instruments were not well-established. In 2012, another systematic review [14] identified two most feasible IPV screening instruments in primary healthcare settings in Afghanistan and Pakistan using the modified versions of the critical appraisal skills pro-gramme criteria not focused on the psychometric properties. However, in 2016, there was a systematic review [15] focused on the psychological properties of IPV screening instruments to screen men and women in mental health settings using the effective public health practice project criteria, which identified ten IPV screening instruments. In 2022, a systematic review [16] focused on risk assessment instruments for IPV not screening IPV instruments, but this systematic review noted implications for prediction and prevention on IPV and gave us future research directions on summarizing risk assessment instruments for IPV. In 2022, there were two important systematic review summarizing the psychometric properties using the COnsensus-based Standards for the selection of health Measurement INstruments (COSMIN) methodology, but one [17] focused on all psychometric properties of dating violence screening instruments among

adolescents and young people between 15 and 24 years of age, another [18] only evaluated reliability and/or validity of included all IPV screening instruments.

Multiple instruments have been developed to screen for IPV, whereas the latest information on their comprehensively methodological quality and psychometric properties have not been assessed. Moreover, the components of IPV screening instruments have not been compared. Researchers and health professionals therefore face challenges regarding which IPV screening instruments to use. Although most IPV screening currently targets women, study [19] still showed that men may also be the victims of IPV. In this study, we hope to explore the methodological properties of IPV screening instruments targeting general population in non-specific population. Therefore, we will include the studies on IPV screening instruments targeting both males and females.

To the best of our knowledge, a systematic review of IPV screening instruments and, specifically, the measurement properties of IPV screening instruments for the general population have not been conducted. The main purpose of this systematic review was therefore to identify the current IPV screening instruments in use for the general population and to evaluate their measurement properties based on the COSMIN methodology for systematic reviews of patient-reported outcome measures (PROMs) [20, 21] and the COSMIN Risk of Bias checklist for systematic reviews of PROMs [22]. We believe that our critical evaluation of the measurement properties of IPV screening instruments will provide useful practical references for other researchers and health professionals who have an interest in IPV among the general population.

## Methods

### Study design

This systematic review was conducted according to the COSMIN guideline for systematic reviews of PROMs and reported in accordance with the PRISMA 2020 checklist [23]. The COSMIN guideline [20] provides methodology on addressing the risk of bias in studies that aimed at instruments developing, rating their measurement properties, evaluating the overall quality of evidence for each measurement property, and providing recommendations based on the overall quality evidence. The methodology of the current study was based on our published protocol (10.3390/ijerph20021541). The protocol [24] for this study was registered in the International Prospective Register of Systematic Reviews (PROSPERO; number CRD42022365247 [24]). Our study did not involve human participants or animals, ethical approval was not applicable.

### Search methods

In March 2024, comprehensively searching in PubMed, Embase, Web of Science, ProQuest Dissertations & Theses Global (PQDT Global), and EBSCO Psychology Behavioral Sciences Collection (PBSC) published databases was done. The Medical Subject Headings (MeSH) terms and 2 keywords in the title (Intimate Partner Violence; Screening Instruments) and abstract were combined with the Boolean operators "AND" and "OR" for the search following a group discussion and assistance from librarians. The search strategies were refined and optimized for each database. An example is shown about the detailed search strategy for the PubMed database in Table 1. The reference lists of review articles identified in the literature search were checked for relevant studies.

**Table 1. An example of the search strategies used in PubMed.**

| # | Search phrase |
|---|---|
| #1 | Intimate partner violence [Mesh] OR Spouse Abuse [Mesh] |
| #2 | Spouse Abuse [T/A] OR Intimate partner violence[T/A] OR IPV[T/A] OR Partner Violence, Intimate[T/A] OR Violence, Intimate Partner[T/A] OR Intimate Partner Abuse[T/A] OR Abuse, Intimate Partner[T/A] OR Partner Abuse, Intimate[T/A] OR Dating Violence[T/A] OR Violence, Dating[T/A] |
| #3 | Mass Screening [Mesh] OR Multiphasic Screening [Mesh] |
| #4 | Screening instrument* [T/A] OR Screening tool* [T/A] |
| #5 | Mass Screening* [T/A] OR Screening*, Mass [T/A] OR Screening* [T/A] |
| #6 | Multiphasic Screening* [T/A] OR Screening*, Multiphasic [T/A] OR Automated Multiphasic Health Testing [T/A] |
| #7 | Screened [T/A] OR detect [T/A] OR detected [T/A] OR detection [T/A] |
| #8 | (#1) OR (#2) |
| #9 | (#3) OR (#4) OR (#5) OR (#6) OR (#7) |
| #10 | (#8) AND (#9) |

## Inclusion and/or exclusion criteria

In this study, we included primary studies that (1) reported screening IPV instruments designed for people in the general population who were victims of IPV, (2) described the processes of development and/or evaluation of one or more measurement properties for the eligible instrument(s), (3) had full-text availability, and (4) articles were published in English. According to the COSMIN guideline, we excluded original studies that used the IPV screening instruments only for outcomes measurements not evaluating measurement properties.

## Study selection

After conducting the search, we used EndNote to manage the references and remove duplicates. We then imported the articles into Joanna Briggs Institute SUMARI search filters to identify the studies with psychometric properties linked to terms related to IPV screening instruments. Two independent researchers (Q.X. and K.C.) undertook the first manual filtering of the articles' titles and abstracts based on the eligibility criteria. The full-text articles not excluded in the second manual screening phase were read independently by two researchers (Y.L. and G.W.). All different opinions between the researchers were resolved under the help of the third researcher (S.O.).

## Data extraction

First, we customized the data extraction table following a discussion in the research group. Then, two researchers (Y.L. and G.W.) separately extracted the corresponding data based on the content of the data extraction table. Finally, the third researcher (J.C.) would examine the extracted data and address any differences encountered. Furthermore, when more than one primary article evaluated the same participants (or the evaluated participants overlapped), information was retrieved from the article that evaluated the larger sample, and the remaining articles were filtered to obtain additional information that was not provided in the main article.

The extracted data included (in Tables 3 and 4): (1) general information about the included studies, such as the publication year and first author's name; (2) the basic characteristics of the identified instruments, including the name of the instrument, original language, target population, country and available translated version, participants and settings, mode of

administration, number of participants (N), item generation, number of items, range of scores, and response options; (3) the results of the measurement properties of the identified instruments, namely, structural validity, internal consistency, cross-cultural validity/measurement invariance, reliability, measurement error, criterion validity, hypothesis testing, and responsiveness; and (4) the feasibility of the instrument, which was not a measurement property but indicated the ease of use of the PROM in its specific context and depended on the acceptability of the research objects, the time and cost to complete the scale, and the scale quality. The feasibility of all the included instruments would be compared to identify the most appropriate screening instruments from those in the same category.

## Quality appraisal and data synthesis

Two researchers (Y.L. and G.W.) independently assessed the measurement properties of the IPV screening instruments using the COSMIN methodology. The third researcher (J.C.) handled any disagreements related to the process. The quality assessment process was divided into three steps: (1) an evaluation of the methodological quality of the included studies; (2) an overall rating of the measurement properties of the included IPV screening instruments; (3) a grading of the quality of the generated evidence. The results of the assessments of the methodological quality of the included studies and the measurement properties of the included IPV screening instruments are shown in Table 5. A detailed description follows.

After first identifying which measurement properties were used in each article, we applied the COSMIN Risk of Bias checklist [22] to assess the methodological quality of the studies included, which encompassed the development of the IPV screening instruments. Each study's quality was assessed independently based on the specific measurement properties, utilizing the corresponding COSMIN box. The quality of each study was rated from "inadequate" to "very good".

Second, we used the updated criteria for good measurement properties to rate the results for the measurement properties of each study. Each result was categorized as either "-" (insufficiency), "?" (indeterminacy), or "+" (sufficiency). Subsequently, the results for each measurement property were aggregated, and the combined or summarized results per measurement property per PROM were re-evaluated utilizing the same criteria. The overall rating of the pooled or summarized results was presented as follows: (1) "+" (sufficient) was given if >75% of the results met the criteria; (2) "-" (insufficient) was given if >75% of the results did not meet the criteria; (3) "±" (inconsistent) was given if no results exceeded 75% and it was not possible to properly explain the inconsistency; and (4) "?" (indeterminate) was given if all the single results were indeterminate.

Third, according to the modified Grading of Recommendations Assessment, Development, and Evaluation approach based on the COSMIN guidelines [20] for the systematic reviews of PROMs, the quality of the evidence was graded from high to very low, which indicated whether the pooled or summarized results were trustworthy, and this was determined by risk of bias, inconsistency, imprecision, and indirectness.

## Recommendations formation

To provide evidenced-based suggestions on the use of IPV screening instruments for researchers, we classified the PROMs into three categories in line with the recommendations in the COSMIN guidelines [20]. Consequently, if a PROM demonstrated sufficient evidence of content validity (at any level) along with at least satisfactory evidence of internal consistency, it was categorized as "A". However, if a measurement property lacked sufficiency despite

possessing high-quality evidence, the PROM was categorized as "C". When the PROM was neither A nor C, the PROM would be classified as "B".

It is noticeable that the PROMs in category A are recommended, but those in category C are not. Of course, the PROMs in category B that have the best content validity could also be used in the interim, but it would be necessary to evaluate the quality of these PROMs further so that better evidence can be provided. Notably, we will give our recommendations comprehensively considering evaluation of the measurement properties and the instrument's feasibility.

# Results

## Search results summary

A total of 9719 records were initially identified through the systematic data search: 2927 in PubMed, 100 in Embase, 6103 in Web of Science, 285 in PQDT Global, and 304 in EBSCO PBSC. After removing the duplicates, screening the titles and abstracts, retrieving the full-text articles, and identifying records from the references of the included studies, 23 studies were eventually included in this systematic review. The process of searching and selection are depicted in Fig 1.

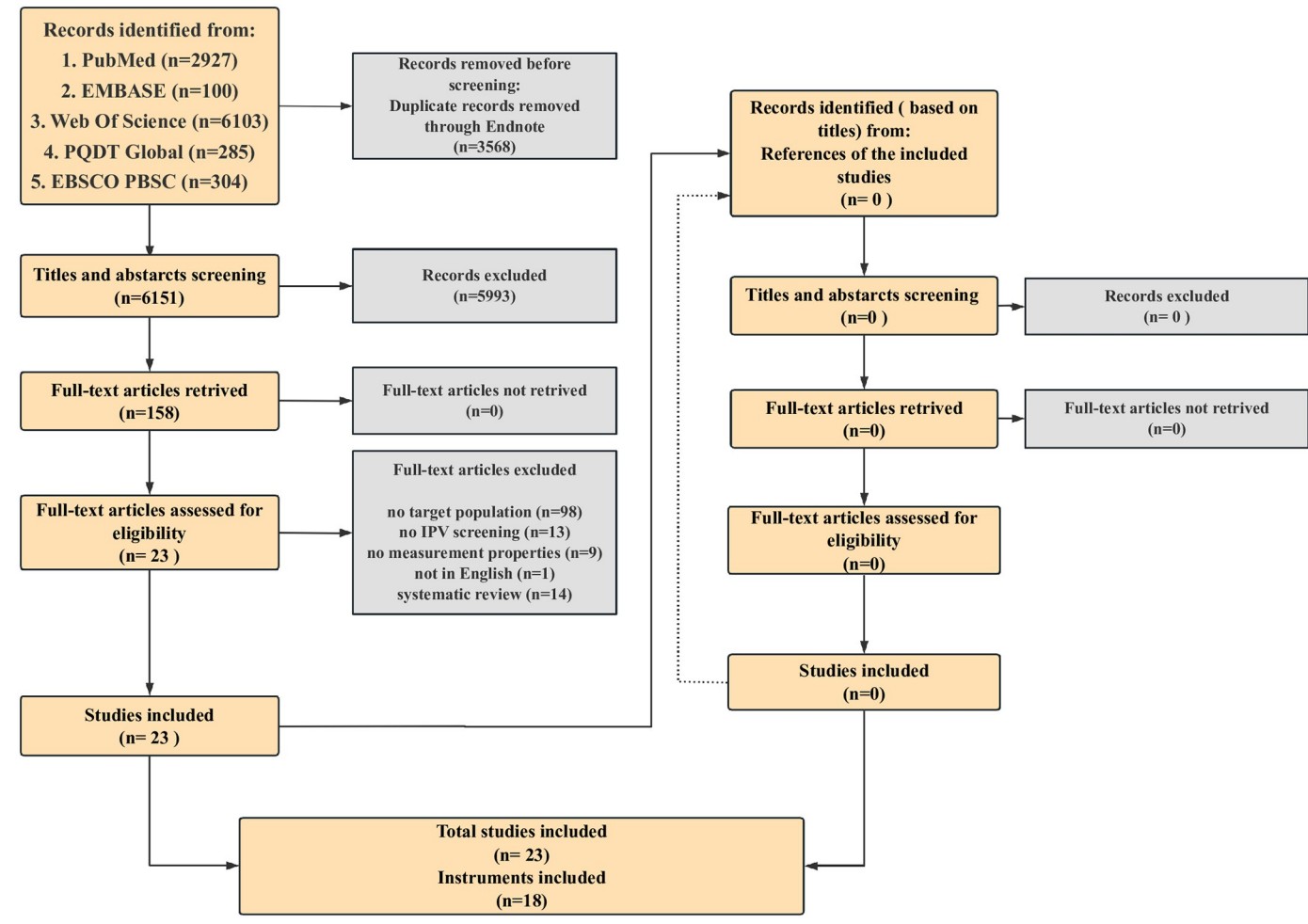

**Fig 1. The process of research searching and selection.**

## Characteristics of the included studies

The included studies were published in English between 1997 and 2022. From these studies, we identified 18 instruments that were specifically developed or translated to screen for IPV in the general population: Partner Violence Screen (PVS) [25–27], the modified version of Partner Violence Screen (M-PVS) [28], the revised Conflict Tactics Scale (CTS-2) [27, 29–31], Abuse Assessment Tool (AAT) [32], Abuse Screening Inventory (ASI) [33], Abuse Behavior Inventory (ABI) [34], the Assessment Screen to Identify Survivors Toolkit for Gender Based Violence (ASIST-GBV) [35], Women's Coerced First Sexual Intercourse (WCFSI) [36], the short form of Sexual Coercion in Intimate Relationships Scale (SCIRS-SF) [37], The South Asian Violence Screen (SAVS) [38], a screening instrument for domestic violence (a screening instrument for DV) [39], a single violence question [40], Haj-Yahia's questionnaire [41], Humiliation-Afraid-Rape-Kick (HARK) [42], NorVold Domestic Abuse Questionnaire (NORAQ) [43], the short form of Psychological Maltreatment of Women Inventory (PMWI-SF) [44], the short version of Multidimensional Scale of Dating Violence (MSDV-2.0) [45], and Intimate Partner Violence—Brief Self-Screener (IBV-BSS) [46]. The detailed information of included studies and instruments is presented in Table 2.

The general information of the included instruments is performed in Tables 3 and 4. Out of the 23 included studies, a total of 18 IPV screening instruments were reported. The 18 instruments were mainly (17 instruments) original developed in English. Among them, M-PVS was

**Table 2. The number of included studies and instruments.**

| Included instrument | | Author, Year |
|---|---|---|
| full name | acronym | |
| Partner Violence Screen | PVS | Harriet 2006 |
| | | Kim 1997 |
| | | Trevor 2005 |
| the modified version of Partner Violence Screen | M-PVS | Nuberg 2008 |
| the revised Conflict Tactics Scale | CTS-2 | Maria 2014 |
| | | Claudia 2002 |
| | | Helen 2015 |
| | | Trevor 2005 |
| Abuse Assessment Tool | AAT | Leila 2006 |
| Abuse Screening Inventory | ASI | KATARINA 2007 |
| Abuse Behavior Inventory | ABI | Zink 2007 |
| the Assessment Screen to Identify Survivors Toolkit for Gender Based Violence | ASIST-GBV | Wirtz 2016 |
| | | Alexander 2016 |
| Women's Coerced First Sexual Intercourse | WCFSI | Shan Shan He 2013 |
| the short form of Sexual Coercion in Intimate Relationships Scale | SCIRS-SF | Guilherme 2021 |
| The South Asian Violence Screen | SAVS | Lenore 2021 |
| a screening instrument for domestic violence | a screen instrument for DV | Taghi 2016 |
| a single violence question | a single violence question | Young-Ju 2017 |
| Haj-Yahia's questionnaire | Haj-Yahia's questionnaire | Sahar 2022 |
| Humiliation-Afraid-Rape-Kick | HARK | Hardip 2007 |
| NorVold Domestic Abuse Questionnaire | NORAQ | Linda 2011 |
| the short form of Psychological Maltreatment of Women Inventory | PMWI-SF | Rita 2018 |
| the short version of Multidimensional Scale of Dating Violence | MSDV-2.0 | Maria 2022 |
| Intimate Partner Violence—Brief Self-Screener | IPV-BSS | Victoria 2021 |

**Table 3. Description of the included IPV screening instruments.**

| Instrument | Categories of Questions | Number of Items | Item generation | Mode of Administration | Response Options | Range of Scores |
|---|---|---|---|---|---|---|
| PVS | ①+safety | 3 | the original version of PVS | self-report | yes or no | not applicable |
| | | 3 | not applicable | | yes or no | not applicable |
| | | 3 | the original version of PVS | | yes or no | not applicable |
| M-PVS | ① | 5 | the original version of PVS | interview | yes or no | not applicable |
| CTS-2 | ①②③④ | 78 | the original version of CTS | self-report | 8-point-likert-type | 78~624 |
| | ①②③④ | 78 | the original version of CTS | | 8-point-likert-type | 78~624 |
| | ①③④ | 32 | the original version of CTS | | 8-point-likert-type | 32~256 |
| | ①②③ | 78 | the original version of CTS | | 7-point-likert-type | 0~468 |
| AAT | ①②③④ | 67 | the finding of a qualitative study and literature review | self-report | 5-point-likert-type | 0~268 |
| ASI | ①②③ | 16 | the texted version of ASI | interview | 11-point-likert-type and yes or no | not applicable |
| ABI | ①③ | 29 | not applicable | self-report | 5-point-likert-type | 29~145 |
| ASIST-GBV | ①②③④ | 7 | the qualitive research and systematic review | self-report | yes or no | not applicable |
| | | 7 | the qualitive research and systematic review | | yes or no | not applicable |
| WCFSI | ①②③④ | 33 | the original version of SCIRS | self-report | 6-point-likert-type | 0~215 |
| SCIRS-SF | ② | 9 | the original version of SCIRS | self-report | 6-point-likert-type | 0~45 |
| SAVS | ①③ | 14 | literature review & the first author's over 20-year experience | self-report | 5-point-likert-type | 14~70 |
| a screen instrument for DV | ①②③ | 20 | HITS & VAWI | self-report | 5-point-likert-type | 0~80 |
| a single violence question | ① | 1 | The partner table | interview | yes or no | not applicable |
| Haj-Yahia's questionnaire | ①②③④ | 32 | CTS+PMWI+MWA+ISA+ABI | self-report | dichotomous scale | not applicable |
| HARK | ①②③④ | 4 | AAS | self-report | yes = 1 & no = 0 | 0~4 |
| NORAQ | | 10 | the English version of NORAQ | self-report | yes or no | not applicable |
| PMWI-SF | ①②③ | 14 | the Portuguese version of PMWI | self-report | 5-point-likert-type | 14~70 |
| MSDV-2.0 | ①②③④ | 42 | the original version of MSDV and new items related to online violence and sexual violence | self-report | 5-point-likert-type | 42~210 |
| IPV-BSS | ①②③④ | 4 | WHO-IPV | self-report | yes or no or not applicable | not applicable |

①physical violence; ② sexual violence; ③emotional-psychological abuse; ④controlling behaviors

translated from English to German, CTS-2 was translated into two languages, namely Italian and Portuguese, WCFSI was translated from English to Chinese, SCIRS-SF was translated from English into Portuguese, a single violence question was translated from Spanish to English, Haj-Yahia's questionnaire was translated from English into Persian, NORAQ was translated from English into Arabic, PMWI-SF was translated from English into Portuguese, and MSDV-2.0 was translated from English into Spanish. The other instruments used a single language (English or Persian) and tested in their target population. The 18 instruments' target

**Table 4. The language and participants of the included IPV screening instruments.**

| Instrument | Original Language | Target Population | Country & Available Translated Version | Participants & Settings | N |
|---|---|---|---|---|---|
| PVS | English | all noncritical English-speaking women | Canada & English | English-speaking women & health care settings | 1602 |
| | | | Denver & English | all noncritical English-speaking women & hospital-based ED | 322 |
| | | | America & English | male victims of IPV & ED | 116 |
| M-PVS | English | women in different institutions from German-speaking countries | Basel & German | German-language women & psychiatric clinic settings | 115 |
| CTS-2 | English | couples in general population | Italy & Italian | non-abused women and victims of IPV & primary health care centers and women's shelters | 209 |
| | | | Rio de Janeiro & Portuguese | pregnant women and premature childbirth & Brazilian context | 774 |
| | | | Victoria & English | separated couples & Family mediation settings | 121 |
| | | | America & English | male victims of IPV & ED | 116 |
| AAT | English | Jamaican women | Jamaica & English | Jamaican women & primary health care clinics and crisis centers | 205 |
| ASI | English | random female Swedish sample | Sweden & English | randomized sample of women & health care settings | 699 |
| ABI | English | English-speaking women | Ohio & English | White, African, and American Women & primary care waiting rooms | 392 |
| ASIST-GBV | English | refugees and internally displaced persons | Ethiopia and Colombia & English | female refugees and internally displaced persons & humanitarian settings | 503 |
| | | | Ethiopia and Colombia & English | | 998 |
| WCFSI | English | intimate relationships | China & Chinese | university students & heterosexual dating settings | 927 |
| SCIRS-SF | English | heterosexual romantic relationships | America and Brazil & Portuguese | Brazilians & heterosexual dating settings | 181 |
| SAVS | English | South Asian immigrant women | Chicago & English | SAI women & most recent relationships | 116 |
| a screen instrument for DV | Persian | Iranian married women | Iran & Persian | married Iranian women & health and social research settings | 334 |
| a single violence question | English | Latina women | Chicago & Spanish or English | Mexican and Puerto-Rican women & community-based healthcare settings | 657 |
| Haj-Yahia's questionnaire | English | female population | Iran & Persian | married women & community-based settings | 471 |
| HARK | English | women in general practice | London & English | women in general practice waiting rooms & clinical settings | 232 |
| NORAQ | English | Arab and Middle Eastern women | Jordan & Arabic | Arab and Middle Eastern women & health -maternal and health care centers | 175 |
| PMWI-SF | English | Portuguese women | Portugal & Portuguese | colleague students & university-based settings | 506 |
| MSDV-2.0 | English | dating relationships | Andalusia & Spanish | university students & educational settings | 1091 |
| IPV-BSS | English | large-scale community populations | Kenya & English | women clients in mental health agencies & clinical and community settings | 8674 |

populations contained women or married women, couples, intimate/romantic/dating relationships, and community populations among general population. Six screening instruments were tested in women and men, and twelve screening instruments only tested in women. Item generation was based on existing IPV screening instruments, literature reviews, reviews of material from service organizations, and/or scholarly work. Fifteen instruments could be administered by self-report, and three instruments required face-to-face interviews. The number of items included in the instruments were diverse and ranged from 1 to 78. The response options were also varied, such as yes or no, 5-point-Likert, or dichotomous.

### Methodological quality and measurement property ratings

The methodological quality and measurement property ratings assessed against the criteria for good measurement properties for each instrument based on the GRADE criteria as well as the PROMs categories are presented in Table 5. The two most frequently reported measurement properties in the included studies were internal consistency and criterion validity. The two least frequently reported measurement properties in the included studies were cross-culture validity and hypotheses testing. Two measurement properties (i.e., measurement error and responsiveness) were not assessed. In terms of methodological quality, the structural validity of the studies in this systematic review was rated as "very good," "adequate," or "doubtful." The most of the studies' internal consistency, reliability, and cross-cultural validity were rated as "very good." The methodological quality of the hypothesis testing was found to be "very good" in only one study, while the methodological quality of the criterion validity of most of the studies was "inadequate."

### Recommendations

We developed our recommendations based on the COSMIN guidelines. We classified screening instruments for domestic violence, Haj-Yahia's questionnaire, NorVold Domestic Abuse Questionnaire, and the short version of Multidimensional Scale of Dating Violence in category "B", so they can be used in the interim until better evidence can be provided. Considering the inadequate methodological of the content validity of the studies, the other 14 instruments are not recommended to use.

## Discussion

In this review, we identified 18 instruments that had been developed to screen for IPV in the general population. To the best of our knowledge, this is the first systematic review to critically evaluate the methodological quality of individual IPV screening instruments and the quality of their measurement properties based on the COSMIN methodology [20, 21]. In this review, four instruments were classified as category "B" and could be used in the interim, and 14 instruments were classified as category "C" and cannot be recommended for use. Further studies are therefore needed to develop instruments that can better screen for IPV in the general population. Although multiple instruments have been used to screen IPV in the general population, many issues still need to be resolved.

A lack of content validity with sufficient methodological quality may affect all the other measurement properties, which represents the relevance, comprehensiveness, and comprehensibility of an instrument [21]. It is essential for an IPV screening instrument to have sufficient content validity, as this contributes to the overall quality of the screening. None of the studies adhered to the minimum requirement recommended by the COSMIN Risk of Bias checklist that 30 experts evaluate the content validity of an instrument [22]. According to many systematic reviews based on COSMIN methods, it is common for content validity to be of a doubtful or inadequate methodological quality [47, 48]. In our opinion, there are two reasons for this phenomenon. First, researchers may overlook the basic criteria and importance of evaluating the content validity of an instrument as per the COSMIN Risk of Bias checklist. Second, the minimum number of experts suggested in the COSMIN Risk of Bias checklist to assess content validity is difficult for researchers to achieve.

The evaluation of structural validity usually uses confirmatory factor analysis and exploratory factor analysis based on the classical test theory. Confirmatory factor analysis is superior to exploratory factor analysis in determining the methodological quality of structural validity according to the COSMIN Risk of Bias checklist [22]. In this systematic review,

**Table 5. Methodological quality and measurement property ratings of included instruments.**

| Instrument | Structural validity n | meth qual | Result (Rating) | Internal consistence n | meth qual | Result (Rating) | Cross-cultural validity n | meth qual | Result (Rating) | Reliability n | meth qual | Result (Rating) | Criterion validity n | meth qual | Result (Rating) | Hypotheses testing n | meth qual | Result (Rating) | Level of evidence for measurement property | PROM category |
|---|---|---|---|---|---|---|---|---|---|---|---|---|---|---|---|---|---|---|---|---|
| IPV-BSS | | | | 544 | red | (+) | | | | | | | | | | | | | High | C |
| SCIRS-SF | 508 | red | (+) | 508 | red | (+) | | | | | | | | | | 508 | red | (-) | Moderate | C |
| SAVS | | | | 116 | red | (+) | | | | | | | 116 | red | (+) | | | | High | C |
| a screen instrument for DV | 334 | red | (+) | 334 | red | (+) | | | | 334 | red | (+) | | | | | | | High | B |
| Haj-Yahia's questionnaire | 471 | red | (+) | 471 | red | (+) | | | | 471 | red | (+) | | | | | | | High | B |
| NORAQ | | | | 171 | red | (+) | | | | | | | | | | | | | High | B |
| PMWI-SF | 506 | yellow | (-) | 506 | red | (+) | | | | 506 | red | (+) | 506 | red | (+) | | | | Moderate | C |
| MSDV-2.0 | 1091 | yellow | (?) | 1091 | red | (+) | | | | 1091 | red | (+) | | | | | | | High | B |
| ABI | | | | 400 | red | (+) | | | | | | | 400 | red | (+) | | | | High | C |
| ASIST-GBV | | | | 503 | red | (+) | 503 | red | (+) | | | | | | | | | | High | C |
| | | | | 998 | red | | 998 | | | | | | | | | | | | | |
| WCFSI | | | | 927 | red | (+) | | | | | | | | | | | | | High | C |
| AAT | | | | 205 | blue | (+) | | | | | | | | | | | | | Low | C |
| ASI | | | | | | | | | | | | | 699 | green | (?) | | | | Very low | C |
| a single violence question | | | | | | | | | | | | | 657 | green | (?) | | | | Very low | C |
| HARK | | | | | | | | | | | | | 232 | green | (?) | | | | Very low | C |
| PVS | | | | | | | | | | | | | 2461 | green | (?) | | | | Very low | C |
| | | | | | | | | | | | | | 322 | | | | | | | |
| | | | | | | | | | | | | | 116 | | | | | | | |
| M-PVS | | | | | | | | | | | | | 115 | green | (?) | | | | Very low | C |
| CTS-2 | 209 | blue | (-) | 209 | red | (+) | | | | 165 | yellow | (+) | 116 | green | (?) | | | | Very low | C |
| | | | | 768 | red | (+) | | | | | | | | | | | | | | |
| | | | | 121 | green | (+) | | | | | | | | | | | | | | |

Red = very good; yellow = adequate, blue = doubtful, green = inadequate

confirmatory factor analysis was used to test the structural validity of five instruments, and confirmatory factor analysis and exploratory factor analysis were used in one [39]. Because no detailed exploratory factor analysis criteria were provided to assess structural validity, we made our final judgements using the confirmatory factor analysis results. Detailed criteria for assessing the results of studies that use exploratory factor analysis should be reported in the COSMIN Risk of Bias checklist in the future [22] so that the methodological quality of instruments can be assessed comprehensively based on the results of confirmatory factor analysis and exploratory factor analysis, and the consistency and comparability of the results can be maintained.

In our review, 11 studies [25, 33, 34, 38, 44] reported the criterion validity of 9 instruments, and the methodological quality of these studies were either very good or inadequate. The common issue regarding the inadequate methodological quality of the criterion validity was that the researchers did not calculate the correlations or the area under the receiver operating curve. Based on the COSMIN guideline, we suggest that researchers select a reasonable "gold standard" to calculate the correlations with the IPV screening instruments they have developed or plot the area under the receiver operating curve to better reflect the criterion validity.

Only one study [37] in our systematic review clearly reported the measurement properties of the comparable instruments when conducting hypothesis testing of the construct validity. When a newly developed instrument is compared to another instrument, the construct of the comparable instrument should be known, and the instrument itself should be of sufficient quality [20]. Future studies should also report the measurement properties of the comparative instruments when developing IPV screening instruments.

Through the above analysis, future studies on developing IPV screening instruments can consider the following points: 1) recognize the importance of content validity and 30 experts evaluate the content validity of an instrument, 2) use confirmatory factor analysis to test structural validity, 3) select an appropriate "gold standard" to calculate the correlations or the area under the receiver operating curve to reflect criteria validity, 4) compare the measurement properties between new IPV screening instruments and other existed IPV screening instruments, and 5) report all measurement properties as much as possible according to the COSMIN guideline, which will benefit the IPV screening instruments elevating from category "B" to "A".

Additionally, the IPV screening instruments in our review could be administered via self-report [45] or face-to-face interviews [28, 40]. We had difficulty identifying which method would be better for screening IPV in the general population. Due to cultural factors, irrespective of whether self-report or face-to-face interviews are used, this population often does not actively disclose their experiences of IPV in their daily lives and even conceal the fact of experiencing IPV. It is noticeable that a cross-sectional study [49] among general population has suggested women reported higher levels of concerns and psychological distress than men. We therefore suggest that future studies using IPV screening instruments should combine self-report and face-to-face interviews to compare the prevalence of IPV among women and men in the general population. The IPV screening instruments in our review had many response options [41, 45, 46], which means there is no unified standard with which to determine the severity of IPV or to measure the cut-off scores of IPV screening instruments. As we mentioned before, because of unwilling disclosing or concealing the experiences of IPV, the results of screening cannot meet the cut-off scores, leading to the prevalence of IPV would be underestimated. In future studies, we hope that researchers will establish a rigorous standard that considers the severity of IPV as well as consistent cut-off scores.

## Limitations

This systematic review had some limitations that need to be acknowledged. First, we only searched five published databases using refined and optimized search strategies, so some articles may have been overlooked. In future studies, more diverse databases should be utilized to ensure a more comprehensive search for articles. Second, we were not able to obtain the measurement errors and responsiveness information from the included instruments. We therefore recommend that, in future studies, researchers examine the measurement properties of the developed IPV screening instruments using the COSMIN methodology as a guideline.

## Implications for future research and clinical practice

This systematic review gives several areas of research on nursing practice that need further exploration. Establishing a unified standard for determining the severity of IPV is essential for consistent assessment and intervention. Further research should focus on developing and validating such standards to ensure accurate identification and classification of IPV cases. Future research is needed to determine cut-off scores for IPV screening instruments, which will enhance the effectiveness of screening protocols by providing clear thresholds for identifying individuals at risk of or experiencing IPV. Futural study is also needed to develop screening instruments on different types of IPV following the COSMIN guideline, which can ensure ensures the reliability, validity, and responsiveness of screening tools across diverse populations and settings.

Implementing standardized assessment for the psychological properties of IPV screening instruments based on COSMIN methodology can help nurses and healthcare professionals to select an appropriate instrument to enhance the identification of IPV in general population. Our findings can inform the development of healthcare policies aimed at addressing IPV. Policymakers can use evidence-based recommendations to advocate for the implementation of comprehensive IPV screening programs and the integration of IPV services into healthcare systems. The safety of general population that experiencing or have experienced IPV will hopefully increase, when these advances are made in the process of healthcare, practice, policy, and research.

## Conclusion

This is the first systematic review to provide a comprehensive overview of the quality of the measurement properties and methodological quality of IPV screening instruments for the general population. Based on the COSMIN guideline, the four instruments in category "B" can be used as interim measures; however, the quality of these PROMs will need to be assessed in future studies so that better evidence can be provided. The overall methodological quality of majority of the evaluated instruments was insufficient. While the methodological quality of the structural validity, internal consistency, and reliability of most of the studies was very good, the methodological quality of the criterion validity and content validity was inadequate. Based on the results of this systematic review, we recommend that nursing and healthcare researchers follow the COSMIN guideline when developing IPV screening instruments. A rigorous IPV screening instrument with good measurement properties is urgently required to identify and screen for IPV in the general population.

## Supporting information

**S1 File. PRISMA 2020 checklist.**
(DOCX)

**S2 File. Search strategies.**
(DOCX)

**S1 Table. The extraction process information.**
(DOCX)

**S2 Table. Boxes of COSMIN risk of bias checklist of the included studies.**
(DOCX)

**S1 Data. Retrieve records.**
(XLSX)

## Acknowledgments

We would like to thank our university librarian Xiaoting Huang for her help with determining the finally refined and optimized search strategies in each database for this study. We also would like to thank the fundings provided support for the design of this study.

## Author Contributions

**Conceptualization:** Siyuan Tang.

**Data curation:** Qing Xia, Keyi Chen, Suqi Ou.

**Formal analysis:** Yanjia Li, Qing Xia, Keyi Chen.

**Funding acquisition:** Jiarui Chen.

**Methodology:** Yanjia Li, Guiyun Wang, Qing Xia, Keyi Chen.

**Project administration:** Jiarui Chen.

**Software:** Yanjia Li, Guiyun Wang, Qing Xia, Keyi Chen.

**Supervision:** Jiarui Chen, Siyuan Tang.

**Validation:** Guiyun Wang, Suqi Ou, Siyuan Tang.

**Visualization:** Suqi Ou.

**Writing – original draft:** Yanjia Li, Guiyun Wang.

**Writing – review & editing:** Jiarui Chen, Suqi Ou, Siyuan Tang.

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
