## [Decision Letter · Decision Letter 0]

24 Jun 2024

PONE-D-24-17901The best intimate partner violence screening instruments for the general population: a COSMIN-based systematic reviewPLOS ONE

Dear Dr. Chen,

Thank you for submitting your manuscript to PLOS ONE. After careful consideration, we feel that it has merit but does not fully meet PLOS ONE’s publication criteria as it currently stands. Therefore, we invite you to submit a revised version of the manuscript that addresses the points raised during the review process.

Although both reviewers found the topic of the study interesting, they also identified numerous methodological shortcomings and a potential limited impact on the international scientific community. The manuscript requires extensive revision and integration of information to ensure its scientific acceptability and suitability for publication.

We look forward to receiving your revised manuscript.

Kind regards,

Stefano Federici, Ph.D.

Academic Editor

PLOS ONE

Journal Requirements:

3. Thank you for stating the following financial disclosure: "This research was funded by the Natural Science Foundation of Hunan Province (grant number 2022JJ40641) and the Changsha Natural Science Foundation (grant number kq2202114), which provided support for the design of this study."

Additional Editor Comments:

Although both reviewers found the topic of the study interesting, they also identified numerous methodological shortcomings and a potential limited impact on the international scientific community. The manuscript requires extensive revision and integration of information to ensure its scientific acceptability and suitability for publication.

Reviewers' comments:

Reviewer's Responses to Questions

**Comments to the Author**

1. Is the manuscript technically sound, and do the data support the conclusions?

Reviewer #1: No

Reviewer #2: Partly

2. Has the statistical analysis been performed appropriately and rigorously? 

Reviewer #1: N/A

Reviewer #2: N/A

3. Have the authors made all data underlying the findings in their manuscript fully available?

Reviewer #1: Yes

Reviewer #2: Yes

4. Is the manuscript presented in an intelligible fashion and written in standard English?

Reviewer #1: No

Reviewer #2: No

5. Review Comments to the Author

Reviewer #1: I believe that the topic is important, however, the authors did not make a strong enough case for why this article is important. Also, the word "best" should be avoided and replaced with a term more appropriate for academic works and audiences. Finally, was this an international/global review (I think it was), and if so, the authors should consider a thorough review of cultural factors related to screening instruments for IPV.

Reviewer #2: General

This paper assumes that the reader knows a lot about psychometrics and COSMIN. Both topics are quite complex and especially as COSMIN is not that well known outside of psychometrics, it would be useful to provide the reader with more information to help them interpret the manuscript. The manuscript would benefit from a table or figure explaining concepts such as construct validity, internal consistency etc and how COSMIN ideally wants these to be measured by studies and what is considered to be less adequate (e.g. that for reliability a study would receive a sufficient rating if the ICC or weighted Kappa ≥ 0.70, the study would be insufficient if the ICC or weighted Kappa was < 0.70 and would be indeterminate is neither ICC nor weighted Kappa was recorded).

I really liked the discussion section of this manuscript, but there should be more information to help the reader before this point.

Abstract

Aim: don’t think need added part of, "for their users". Recommendations is fine.

PQDT and EBSCO PBSC should be in full, not the acronym.

Introduction

Line 46: What do you mean by other relationship types? I would assume it’s some form of romantic/dating or sexual relationship? This needs to be made clear.

Line 47 – 49: The sentence about domestic violence isn’t as easy to understand as it could be. Suggest that you explain that you’re going to refer to the concept as intimate partner violence, rather than domestic violence as you’re only looking at instruments that address abuse between romantic/dating partners rather than abuse between family members (e.g. abuse of children or elderly relatives).

Line 52 – 57: When you say the physical health consequences of IPV are well documented and the mental health consequences are frequently documented, what do you mean? Are there differences between the amount of evidence available for physical and mental health? Are associations between IPV and physical health better established? Or are the links between IPV and physical and mental health equally well evidenced? The wording of this part of the manuscript implies a hierarchy of evidence, but I'm not sure if that's what you meant.

Line 63: "Because of the difficulty of meeting the cut-offs" – I’m not sure what you mean by this. This should be clearly explained.

Line 89: I wouldn’t give the example of female patients. State that you include studies on IPV screening instruments targeting both males and females.

Some of the phrasing in the introduction needs a re-think. It can be quite difficult to follow.

Methods

Provide references for COSMIN.

COSMIN addresses the psychometric properties of measurement instruments. The risk of bias tool is an adaptation of the checklist and its four-point rating system into a version exclusively for use in systematic reviews of Patient Reported Outcome Measures and assesses risk of bias of studies on measurement properties – Mokkink et al (2018) COSMIN Risk of Bias checklist for systematic reviews of Patient-Reported Outcome Measures. Qual Life Res. It’s not specifically about IPV screening instruments, as could be interpreted here.

Line 114/115 – why did you contact the author of each included study to review the search results?

Inclusion/Exclusion criteria isn't detailed enough, use PICO (population, interventions, comparators and outcomes).

I would be more inclined to say people who are victims/survivors of IPV rather than people who were suffering from IPV.

Were reference lists of review articles identiﬁed in the literature search checked for relevant studies?

Were there exclusions for language?

Study selection

Examples of the Joanna Briggs Institute SUMARI search filters used would be helpful for someone replicating the review.

Who reviewed the titles and abstracts?

Data abstraction

This is more of a computer science term, I would title as data extraction.

PROM needs writing out in full the first time it’s mentioned.

Characteristics of the included studies

Were the instruments tested mainly with males or females? or a mixture of both?

Methodological quality and measurement property ratings

What were the psychometric properties that were reported the least in the studies?

More information is needed to explain why the measurement properties were given the ratings that they were. What was presented in the studies for the internal consistency, reliability and cross-cultural validity to be rated as very good? What was missing for criterion validity to be rated inadequate?

Should be clearer how many studies measured each psychometric property for each instrument, I assume with 18 instruments and 23 studies, most instruments will have only been assessed by one study, but this isn’t clear. A table, figure or some reporting would help with this. Readers should understand this before the discussion.

Discussion

Some discussion of what evidence is currently missing that would elevate existing measures from a B to an A may be interesting, especially with regards to directions for future research.

Generally the discussion section is very well written. My only comment is that the rest of the paper refers to IPV among both men and women, but the discussion only mentions women being reluctant to report.

The discussion would benefit from some inclusion about the majority of instruments only being tested with women.

Table 2

Table 2 may benefit from more detail about the thematic content of the instruments. For example, categories of questions. Alternatively, this could be a supplementary table.

Table 2 and 3

The acronym and full name of each instrument would be useful here, so the reader don't have to look back to the main text.

Table 3

Make it clear the levels of evidence is using the GRADE criteria.

There are spelling mistakes in the figure presented.

6. PLOS authors have the option to publish the peer review history of their article (what does this mean?). If published, this will include your full peer review and any attached files.

Reviewer #1: No

Reviewer #2: No

---

## [Author Response · Author response to Decision Letter 0]

13 Aug 2024

Dear Editor-in-Chief and reviewers，

Thank you for the quick and positive feedback for the manuscript. We sincerely thank the editor and reviewers for their constructive comments and suggestions on further revisions to improve the quality of this manuscript. Appended to this letter is our point-by-point responses to the comments and suggestions. Changes made in the manuscript are marked in red in the uploaded files. A clean and a track-change copies of the revised manuscript and supporting information files have been uploaded. The response to reviewers have been uploaded. 

We hope that you find our responses satisfactory, and that the manuscript is now acceptable for publication in the PLOS ONE. 

Yours sincerely,

Jiarui CHEN

Xiangya School of Nursing,

Central South University

Reviewer 1:

1. I believe that the topic is important, however, the authors did not make a strong enough case for why this article is important. Also, the word "best" should be avoided and replaced with a term more appropriate for academic works and audiences. Finally, was this an international/global review (I think it was), and if so, the authors should consider a thorough review of cultural factors related to screening instruments for IPV.

Response: Thank you very much for your comments. We have noticed that using “best” is not appropriate in our manuscript. The title revision is as follows: “Evaluation of the measurement properties of intimate partner violence screening instruments for the general population: a COSMIN-based international systematic review”. -Page 1 lines 1-2.

Moreover, for culture factors, we added corresponding contents in the part of results. The revision is as follows: “Out of the 23 included studies, a total of 18 IPV screening instruments were reported. The 18 instruments were mainly (17 instruments) original developed in English. Among them, M-PVS was translated from English to German, CTS-2 was translated into two languages, namely Italian and Portuguese, WCFSI was translated from English to Chinese, SCIRS-SF was translated from English into Portuguese, a single violence question was translated from Spanish to English, Haj-Yahia's questionnaire was translated from English into Persian, NORAQ was translated from English into Arabic, PMWI-SF was translated from English into Portuguese, and MSDV-2.0 was translated from English into Spanish. The other instruments used a single language (English or Persian) and tested in their target population. The 18 instruments’ target populations contained women or married women, couples, intimate/romantic/dating relationships, and community populations among general population.” -Page 11 to 12 lines 203-213.

We consider that the translated IPV screening instruments could be tested in corresponding participants different from their target population, which could proof this IPV screening instruments adopted their culture.

Reviewer 2:

1. This paper assumes that the reader knows a lot about psychometrics and COSMIN. Both topics are quite complex and especially as COSMIN is not that well known outside of psychometrics, it would be useful to provide the reader with more information to help them interpret the manuscript. The manuscript would benefit from a table or figure explaining concepts such as construct validity, internal consistency etc. and how COSMIN ideally wants these to be measured by studies and what is considered to be less adequate (e.g. that for reliability a study would receive a sufficient rating if the ICC or weighted Kappa ≥ 0.70, the study would be insufficient if the ICC or weighted Kappa was < 0.70 and would be indeterminate is neither ICC nor weighted Kappa was recorded).

Response: Thank you for your opinions. After your comments, we have noticed that we haven’t add the published protocol of this review as a reference, Actually, before we conducted the full review, we have published a protocol titled on “Intimate Partner Violence Screening Instruments: A Protocol for a COSMIN-Based Systematic Review” (DOI: 10.3390/ijerph20021541). Therefore, the methodology of our study was based on this protocol, the contents of these concepts such as construct validity, internal consistency etc. and how COSMIN ideally wants these to be measured by studies and what is less adequate were not repeated in our manuscript. We have added corresponding explanations in the method part as follows: “The methodology of the current study was based on our published protocol (DOI: 10.3390/ijerph20021541). The protocol (24) for this study was registered in the International Prospective Register of Systematic Reviews (PROSPERO; number CRD42022365247 (24)).” –Page 5 lines 98-100. 

2. Abstract

Aim: don’t think need added part of, "for their users". Recommendations is fine.

PQDT and EBSCO PBSC should be in full, not the acronym. 

Response: Thanks for your suggestions. The revisions are as follows: “Aim: To systematically appraise, compare, and summarize the measurement properties of intimate partner violence screening instruments for the general population and provide recommendations based on COSMIN guideline. 

Methods: We searched PubMed, Embase, Web of Science, ProQuest Dissertations & Theses Global and EBSCO Psychology Behavioral Sciences Collection from their establishment to March 2024 using systematic search strategies.” -Page 2 lines 24-28.

3. Introduction

Line 46: What do you mean by other relationship types? I would assume it’s some form of romantic/dating or sexual relationship? This needs to be made clear.

Response: Thank you very much for your suggestions. According to your suggestions, we read again the literature. We apologized for the confusions. The revision is as follows: “ Intimate partners may include couples, ex-couples, boyfriends/girlfriends, dating partners, sexual partners, or other romantic type of relationship.” -Page 3 lines 46-47.

4. Line 47 – 49: The sentence about domestic violence isn’t as easy to understand as it could be. Suggest that you explain that you’re going to refer to the concept as intimate partner violence, rather than domestic violence as you’re only looking at instruments that address abuse between romantic/dating partners rather than abuse between family members (e.g. abuse of children or elderly relatives).

Response: Thank you for your comments, we apologized for the confusions. The revision is as follows: “The term “domestic violence” is a broader term that contains the abuse of children or the elderly or the abuse by any member from the family, while it was used in many countries to refer to intimate partner violence (5)” -Page 3 lines 48-50.

5. Line 52 – 57: When you say the physical health consequences of IPV are well documented and the mental health consequences are frequently documented, what do you mean? Are there differences between the amount of evidence available for physical and mental health? Are associations between IPV and physical health better established? Or are the links between IPV and physical and mental health equally well evidenced? The wording of this part of the manuscript implies a hierarchy of evidence, but I'm not sure if that's what you meant.

Response: Thanks for your comments. After group discussion, we recognized that you are right. The revision is as follows: In a study by the World Health Organization, 27% of women aged 15–49 years had been subjected to physical and/or sexual violence by their partners, and 13% of them had experienced such events in the previous 12-month period (7). IPV can result in negative physical and emotional outcomes for the people who experience violence and their families (2). The physical health consequences include acute illnesses and chronic and pain-based disorders, such as respiratory, urinary tract, and sexually transmitted illnesses, insomnia, headache, menstrual-related disorders, pelvic pain, and functional gastrointestinal and reproductive diseases (8-10). Mental health issues are also reported among people who are experiencing or have experienced IPV, and these can have potentially serious consequences. Anxiety, depression, posttraumatic stress disorder, and suicide are all common among this population (11, 12). Importantly, the severity of these mental health issues tends to escalate with the severity of the IPV experienced.” -Page 3 lines 52-61.

6. Line 63: "Because of the difficulty of meeting the cut-offs" – I’m not sure what you mean by this. This should be clearly explained.

Response: Thanks for your comments. We have added the explanation in the part of discussion in our manuscript, the added contents are as follows: “ The IPV screening instruments in our review had many response options (41, 45, 46), which means there is no unified standard with which to determine the severity of IPV or to measure the cut-off scores of IPV screening instruments. As we mentioned before, because of unwilling disclosing or concealing the experiences of IPV, the results of screening cannot meet the cut-off scores, leading to the prevalence of IPV would be underestimated. In future studies, we hope that researchers will establish a rigorous standard that considers the severity of IPV as well as consistent cut-off scores.” -Page 21 lines 332-337.

7. Line 89: I wouldn’t give the example of female patients. State that you include studies on IPV screening instruments targeting both males and females.

Response: Thanks for your comments. We have noticed our expression was described unclearly. The revisions are as follows: “In this study, we hope to explore the methodological properties of IPV screening instruments targeting general population in non-specific population. Therefore, we will include the studies on IPV screening instruments targeting both males and females.” -Page 4 lines 81-83.

8. Methods

Provide references for COSMIN.

COSMIN addresses the psychometric properties of measurement instruments. The risk of bias tool is an adaptation of the checklist and its four-point rating system into a version exclusively for use in systematic reviews of Patient Reported Outcome Measures and assesses risk of bias of studies on measurement properties – Mokkink et al (2018) COSMIN Risk of Bias checklist for systematic reviews of Patient-Reported Outcome Measures. Qual Life Res. It’s not specifically about IPV screening instruments, as could be interpreted here.

Response: Thank you very much for your suggestion. The revised content is as follows: “The COSMIN guideline (20) provides methodology on addressing the risk of bias in studies that aimed at instruments developing, rating their measurement properties, evaluating the overall quality of evidence for each measurement property, and providing recommendations based on the overall quality evidence. The protocol for the current study was registered in the International Prospective Register of Systematic Reviews (PROSPERO; number CRD42022365247).” -Page 5 lines 95-98.

9. Line 114/115 – why did you contact the author of each included study to review the search results? Were reference lists of review articles identiﬁed in the literature search checked for relevant studies?

Response: Thanks for your question. We apologized for the expression unclear. The revised content is as follows: “ The reference lists of review articles identiﬁed in the literature search were checked for relevant studies. ” -Page 5 line 108.

10. Inclusion/Exclusion criteria isn't detailed enough, use PICO (population, interventions, comparators and outcomes).

I would be more inclined to say people who are victims/survivors of IPV rather than people who were suffering from IPV.

Were there exclusions for language?

Response: Thanks for your suggestions. Because we don’t cite our published protocol in our manuscript, we apologized for the confusions. The contents on PICO in our protocol are as follows: “In this study, we adapted the ‘objectives’ section of PRISMA 2020 checklist by changing ‘Population(s), Interventions, Comparators and Outcomes’ into ‘Construct, Populations, Type of Instruments and Measurement Properties’. The main search concepts are the consequences of IPV (Construct), people who suffer from IPV (Population), PROMs (Type of instrument(s)), and measurement properties.” 

Additionally, the questions for languages we wrote in the inclusion criteria, the revisions are as follows: “Therefore, we included primary studies that (1) reported screening IPV instruments designed for people in the general population who were victims of IPV, (2) described the processes of development and/or evaluation of one or more measurement properties for the eligible instrument(s), (3) had full-text availability, and (4) articles were published in English. According to the COSMIN guideline, we excluded original studies that used the IPV screening instruments only for outcomes measurements not evaluating measurement properties.” -Page 6 lines 112-116.

11. Study selection

Examples of the Joanna Briggs Institute SUMARI search filters used would be helpful for someone replicating the review.

Response: Thanks for your suggestions. We used the Joanna Briggs Institute SUMARI search filter in our study, which was helpful for us. We wrote in our manuscript; the content is as follows: “We then imported the articles into Joanna Briggs Institute SUMARI search filters to identify the studies with psychometric properties linked to terms related to IPV screening instruments.” -Page 7 lines 118-120.

12. Who reviewed the titles and abstracts?

Response: Thanks for your question. We have added the researchers who reviewed the titles and abstracts in our manuscripts, and the researchers who extracted data and conducted quality appraisal and data synthesis, the revisions are as follows: “ Two independent researchers (Q.X. and K.C.) undertook the first manual filtering of the articles’ titles and abstracts based on the eligibility criteria. The full-text articles not excluded in the second manual screening phase were read independently by two researchers (Y.L. and G.W.). All different opinions between the researchers were resolved under the help of the third researcher (S.O.).” -Page 7 lines 120-123.

“First, we customized the data extraction table following a discussion in the research group. Then, two researchers (Y.L. and G.W.) separately extracted the corresponding data based on the content of the data extraction table. Finally, the third researcher (J.C.) would examine the extracted data and address any differences encountered.” -Page 7 lines 125-127.

“Two researchers (Y.L. and G.W.) independently assessed the measurement properties of the IPV screening instruments using the COSMIN methodology. The third researcher (J.C.) handled any disagreements related to the process.” -Page 8 lines 143-145.

13. Data abstraction

This is more of a computer science term, I would title as data extraction.

PROM needs writing out in full the first time it’s mentioned.

Response: Thanks for your opinions. The title revision is as follows: “ Data extraction”-Page 7 line 124. Additionally, the first time we mentioned PROMs, we have written out in full in the introduction, the content is as follows: “ The main purpose of this systematic review was therefore to identify the current IPV screening instruments in use for the general population and to evaluate their measurement properties based on the COSMIN methodology for systematic reviews of patient-reported outcome measures (PROMs) (20, 21) and the COSMIN Risk of Bias checklist for systematic reviews of PROMs (22).” -Pages 4 to 5 lines 85-89.

14. Characteristics of the included studies

Were the instruments tested mainly with males or females? or a mixture of both?

Response: Thanks for your questions. We have noticed that our expression was unclear, the revision is as follows: “Six screening instruments were tested in women and men, and twelve screening instruments only tested in women.” -Page 12 lines 212-213.

15. Methodological quality and measurement property ratings

What were the psychometric properties that were reported the least in the studies?

Response: Thanks for your questions. We have added the content in our manuscript that is as follows: “The two least frequently reported measurement properties in the included studies 

---

## [Editor Report · Decision Letter 1]

29 Aug 2024

Evaluation of the measurement properties of intimate partner violence screening instruments for the general population: a COSMIN-based international systematic review

PONE-D-24-17901R1

Dear Dr. Chen,

We’re pleased to inform you that your manuscript has been judged scientifically suitable for publication and will be formally accepted for publication once it meets all outstanding technical requirements.

Kind regards,

Stefano Federici, Ph.D.

Academic Editor

PLOS ONE

Additional Editor Comments (optional):

Although both reviewers were invited to comment on the original version, they were unable to participate in a second round for personal reasons. Having carefully reviewed the authors' responses to the reviewers and the changes made to the text, it is my assessment that the manuscript has been significantly enhanced and is now suitable for publication.
---

## [Editor Report · Acceptance letter]

25 Sep 2024

PONE-D-24-17901R1 

PLOS ONE

Dear Dr. Chen, 

I'm pleased to inform you that your manuscript has been deemed suitable for publication in PLOS ONE. Congratulations! Your manuscript is now being handed over to our production team.

Kind regards, 

on behalf of

Prof. Stefano Federici 

Academic Editor

PLOS ONE